# Observation-Based Attitude Realization for Accurate Jason Satellite Orbits and Its Impact on Geodetic and Altimetry Results

**Mathis Bloßfeld** *,† , **Julian Zeitlhöfler** , **Sergei Rudenko** and **Denise Dettmering**

Deutsches Geodätisches Forschungsinstitut, Technische Universität München (DGFI-TUM), 80333 Munich, Germany; julian.zeitlhoefler@tum.de (J.Z.); sergei.rudenko@tum.de (S.R.); denise.dettmering@tum.de (D.D.)

*   Correspondence: mathis.blossfeld@tum.de; Tel.: +49-89-23031-1119
†   Current address: Arcisstr. 21, 80333 Munich, Germany.

**Abstract:** For low Earth orbiting satellites, non-gravitational forces cause one of the largest perturbing accelerations. During a precise orbit determination (POD), the accurate modeling of the satellite-body attitude and solar panel orientation is important since the satellite's effective cross-sectional area is directly related to the perturbing acceleration. Moreover, the position of tracking instruments that are mounted on the satellite body are affected by the satellite attitude. For satellites like Jason-1/-2/-3, attitude information is available in two forms—as a so-called nominal yaw steering model and as observation-based (measured by star tracking cameras) quaternions of the spacecraft body orientation and rotation angles of the solar arrays. In this study, we have developed a preprocessing procedure for publicly available satellite attitude information. We computed orbits based on Satellite Laser Ranging (SLR) observations to the Jason satellites at an overall time interval of approximately 25 years, using each of the two satellite attitude representations. Based on the analysis of the orbits, we investigate the influence of using preprocessed observation-based attitude in contrast to using a nominal yaw steering model for the POD. About 75% of all orbital arcs calculated with the observation-based satellite attitude data result in a smaller root mean square (RMS) of residuals. More precisely, the resulting orbits show an improvement in the overall mission RMS of SLR observation residuals of 5.93% (Jason-1), 8.27% (Jason-2) and 4.51% (Jason-3) compared to the nominal attitude realization. Besides the satellite orbits, also the estimated station coordinates benefit from the refined attitude handling, that is, the station repeatability is clearly improved at the draconitic period. Moreover, altimetry analysis indicates a clear improvement of the single-satellite crossover differences (6%, 15%, and 16% reduction of the mean of absolute differences and 1.2%, 2.7%, and 1.3% of their standard deviations for Jason-1/-2/-3, respectively). On request, the preprocessed attitude data are available.

**Keywords:** satellite attitude; precise orbit determination; POD; Jason satellites; satellite laser ranging; SLR; nominal yaw steering model; quaternion; altimetry

## 1. Introduction

Artificial near-Earth satellites are nowadays an effective instrument for Earth observations and the monitoring of global change phenomena. Altimetry satellites provide a continuous data record of global and regional sea level change, see, for example, Reference [1]. Data from these satellites also allow the investigation of ocean dynamics (large- and small-scale circulation, ocean tides, waves, El Niño-Southern Oscillation, coastal processes, etc.), the cryosphere, dynamics of land surface waters, and seafloor topography [2]. For these applications, a highly accurate (at the 1 cm level) and stable

(below 1 mm/year) satellite orbit is required [3,4]. The accuracy and stability of altimetry satellite orbits significantly improved in recent decades due to enhancements in, amongst others, modeling the Earth's time-variable gravity field [5], reference frame determination [6,7], and improvements in modeling non-gravitational perturbations [8,9].

Altimetry satellites such as Jason-1/-2/-3 have a non-spherical, complex shape comprising the main satellite body on which solar panels and numerous measurement and positioning instruments are mounted. A detailed overview of the Jason satellite missions goals, spacecraft and payloads is given in Reference [10] and, for example, the websites of the Earth Observation portal (https://directory.eoportal.org/web/eoportal/satellite-missions/j/jason-2, Last access: 14 November 2019 and https://directory.eoportal.org/web/eoportal/satellite-missions/j/jason-3, Last access: 14 November 2019) and the International DORIS Service (IDS; ftp://ftp.ids-doris.org/pub/ids/satellites/MassCoGInitialValues.txt, Last access: 14 November 2019). Precise information on the size, the optical properties and the orientation in space of the satellite surfaces is important since non-gravitational forces caused by, for example, solar radiation pressure, Earth's reflected and infrared radiation and atmospheric drag depend on the satellite effective cross-sectional area. In addition, the phase center locations of positioning devices such as a laser ranging array (LRA), a Doppler Orbitography and Radiopositioning Integrated by Satellite (DORIS) receiver and a Global Navigation Satellite Systems (GNSS) receiver required in the inertial reference frame depend on the satellite orientation in space. Therefore, so called nominal yaw steering models (see also Section 2.1.1) are defined and can be used to realize a nominal (not ideal) orientation of a spacecraft. In addition to these models, for satellites equipped with star tracking cameras like the Jason satellites, the measured spacecraft orientation is available (see Section 2.1.2). Until recently, mainly a given functional model (nominal law) has been used for the attitude modeling of the Jason satellites. Thus, five out of six Analysis Centers (ACs) of the International DORIS Service (IDS) used the nominal law to compute the satellite attitude when they computed their contribution to the International Terrestrial Reference Frame realization ITRF2014 (https://ids-doris.org/analysis-coordination/combination/contributions-to-itrf/contribution-itrf2014.html, Last access: 16 January 2020). Solely the IDS AC at NASA/GSFC (National Aeronautics and Space Administration/Goddard Space Flight Center) [11] used attitude observations given as satellite body quaternions and solar panel rotation angles for their contribution. Moreover, also the Jason orbits of the Helmholtz-Zentrum Potsdam – Deutsches GeoForschungsZentrum (GFZ, Germany) and the Centre National d'Études Spatiales (CNES, France), which are used for the orbit comparison in Section 3.3, were derived using an observation-based attitude. Another example of the use of observation-based attitude information can be found for the CryoSat-2 satellite [12].

In this paper, we investigate the impact of the observation-based (measured) spacecraft attitude realization in comparison to the modeled nominal yaw steering attitude for the three altimetry satellites Jason-1/-2/-3 [13]. We compare the root mean square (RMS) fits of Satellite Laser Ranging (SLR) observations and quantify the impact on estimated parameters such as scaling factors for the solar radiation pressure, Earth's albedo and thermospheric drag. Empirical accelerations in the along-track and cross-track directions are studied as well as station coordinate time series. Additionally, the derived orbits are compared with selected external orbits and, finally, the RMS and the mean of single-satellite altimetry crossover differences and radial errors are evaluated. This analysis allows the investigation of the impact of the satellite attitude modeling on geographically correlated mean errors (GCEs). In this paper, we use the word "mean" to abbreviate the "arithmetic mean".

The paper is organized as follows. The first part of Section 2 provides information on two satellite attitude modeling strategies, namely, the nominal yaw steering model and the observation-based attitude realization. The second part of Section 2 describes the models used for orbit determination for these satellites. The POD and altimetry results of these tests are described in Section 3. Finally, Section 4 discusses the obtained results and concludes the paper.

## 2. Methods and Materials

### 2.1. Satellite Attitude Modeling

For the POD of a satellite orbiting the Earth in the Geocentric Celestial Reference System (GCRS) based on observations of stations given in the International Terrestrial Reference System (ITRS), multiple rotations are involved. The transformation between the GCRS and the ITRS is realized by so called Earth Orientation Parameters (EOP). The transformation between the GCRS and the satellite body reference system (SAT), also called the satellite attitude, is realized by

$$
\begin{aligned}
\boldsymbol{X}_{SAT} \;&=\; \boldsymbol{R}_{GCRS}^{SAT} \;\cdot\; \boldsymbol{X}_{GCRS} \\
&=\; \overbrace{\boldsymbol{R}_{ORB}^{SAT} \quad \boldsymbol{R}_{GCRS}^{ORB}} \;\cdot\; \boldsymbol{X}_{GCRS} \\
&=\; \overbrace{\boldsymbol{R}_{RPY}^{SAT} \, \boldsymbol{R}_{ORB}^{RPY}} \, \boldsymbol{R}_{GCRS}^{ORB} \;\cdot\; \boldsymbol{X}_{GCRS},
\end{aligned}
\tag{1}
$$

with $\boldsymbol{X}_{SAT}$ being the satellite position vector in the satellite body reference system and $\boldsymbol{X}_{GCRS}$ being the satellite position vector in the GCRS. Moreover, the following rotation matrices are used:

$\boldsymbol{R}_{GCRS}^{SAT}$　rotation matrix from the GCRS to the satellite body reference system,

$\boldsymbol{R}_{ORB}^{SAT}$　rotation matrix from the orbital reference system to the satellite body reference system,

$\boldsymbol{R}_{RPY}^{SAT}$　rotation matrix from the local orbital (roll-pitch-yaw) to the satellite body reference system (roll and pitch angles account for the differences between geodetic and geocentric pointing),

$\boldsymbol{R}_{ORB}^{RPY}$　rotation matrix from the orbital to the local orbital reference system,

$\boldsymbol{R}_{GCRS}^{ORB}$　rotation matrix from the GCRS to the orbital reference system.

For each matrix, the starting system is denoted in the subscript and the target system in the superscript. It should be mentioned that all coordinate systems involved are right-handed, only passive rotations are applied and the rotation angles are positive in the counter-clockwise direction. In Equation (1), $\boldsymbol{R}_{GCRS}^{SAT}$ can be computed using the quaternions (see Section 2.1.2) whereas the nominal yaw steering model is used for the computation of $\boldsymbol{R}_{RPY}^{SAT}$ (see next Section).

#### 2.1.1. Nominal Yaw Steering Model

The spacecraft attitude of three-axis stabilized altimetry satellites has to comply with two requirements: a constant nadir (Earth-pointing) orientation of the altimeter boresight and simultaneously a solar-pointing orientation of the solar arrays. The first prerequisite is necessary to provide altimeter measurements of the ocean surface. The nadir of the satellite is represented by the Z axis in the satellite body reference system and the yaw axis in the local orbital system (hereafter only the yaw axis is mentioned). Consequently, this axis is steadily normal to the reference ellipsoid. This so-called geodetic pointing is realized by small roll and pitch angles. To ensure a continuous nadir-pointing, large rotations are only allowed around the yaw axis. The second requirement of the continuous Sun-pointing of the solar arrays provides optimal power supply for the satellite payload. The normal of each solar panel should continuously point towards the Sun although the panels can only rotate around an axis which is parallel to the Y axis of the satellite body system.

To maintain both orientations, a yaw steering algorithm and a solar array pitching algorithm were created by Perrygo [14] for Ocean TOPography EXperiment (TOPEX)/Poseidon. The attitude regime of Jason satellites is based on these algorithms. The fundamental parameters of the algorithms are $\beta'$ and $\nu$. The first variable is the angle between the geocentric position vector of the satellite (vector in the orbital plane) and the direction from the Earth's geocenter to the Sun's center of gravity (vector in the ecliptic), that is, the elevation of the Sun w.r.t. the orbital plane, see Figure 1a). The main period of this angle is approximately 117.53 days (Jason draconitic period). The second parameter $\nu$ is the angle between the Earth-Sun vector projected into the orbital plane and the Earth-satellite vector counted in counter-clockwise direction.

The ideal rotation angle $\psi$ around the yaw axis is defined by the function

$$yaw_{ideal} = \psi_{ideal} = \arctan \frac{\tan \beta'}{\sin \nu}. \tag{2}$$

Large values of $\beta'$ imply a small amplitude of the yaw angle and cause a slow change of the spacecraft attitude. Small $\beta'$ angles affect a fast variation of the orientation to maintain the Sun-pointing direction of the solar arrays. Based on the ideal orientation and the angle $\beta'$, the nominal yaw steering algorithm comprises four yaw modes (Table 1). A sinusoidal yaw steering regime is used when $|\beta'| > \beta'_{ramp}$. The $\beta'_{ramp}$ angle is equal to $15°$ for the entire Jason-1 mission, the part of the Jason-2 mission before 14 July 2017 and the part of the Jason-3 mission before 12 August 2017. The $\beta'_{ramp}$ angle was changed by the space agency from $15°$ to $30°$ for Jason-2 and Jason-3 on the dates given, respectively. In this regime the satellite performs an oscillating rotation around the yaw axis (see Equation (3)). However, the sinusoidal yaw regime is an approximation of the ideal orientation which results in an offset between both modes. The maximum difference of approximately $28°$ occurs at $|\beta'| = 15°$ (Figure 1b). The nominal yaw angle for the sinusoidal law regime is computed using the following formula:

$$yaw_{nominal} = \psi_{nominal} = \begin{cases} 90° - (90° - \beta') \sin \nu & \text{, if } \beta' > \beta'_{ramp} \\ -90° + (90° + \beta') \sin \nu & \text{, if } \beta' < -\beta'_{ramp} \end{cases}. \tag{3}$$

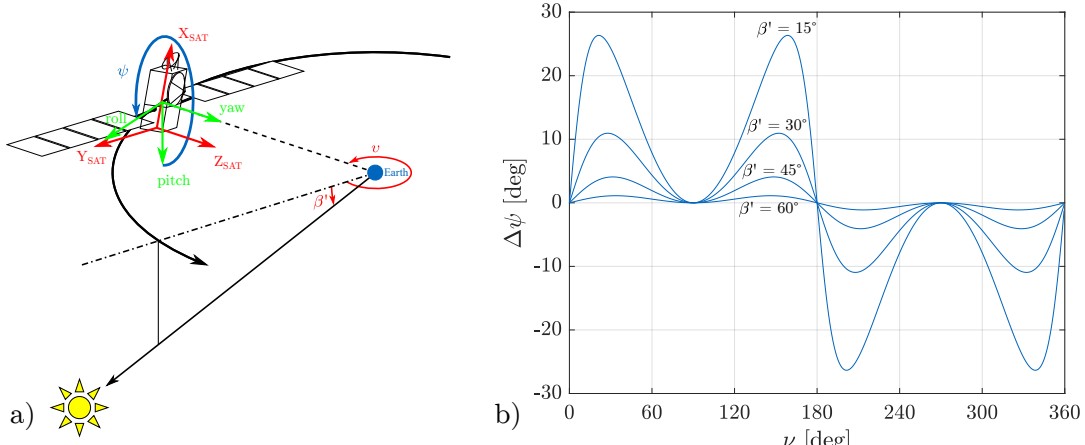

**Figure 1.** (**a**) Definition of the parameters $\beta'$ and $\nu$ (modified from [15]). (**b**) Difference between the nominal and ideal yaw depending on the angle $\beta'$ (modified from [15]).

To avoid excessive rotations, the yaw axis is fixed when $|\beta'| < \beta'_{ramp}$. The yaw angle is set to $\psi = 0°$ at positive $\beta'$ values (spacecraft is flying forward) and to $\psi = 180°$ at negative values (flying backward), respectively. Consequently, a yaw-flip event at approximately $\beta' = 0°$ has to be performed within the fixed-yaw regime. This is required to provide thermal control of sensitive payload instruments and to protect coolers from direct solar illumination. The approximately 95–130 s transition

from the fixed to the sinusoidal yaw regime and vice versa are called ramp-up and ramp-down events, respectively. Both events do not necessarily start at exactly $|\beta'| = \beta'_{\mathrm{ramp}}$ but can differ by some degrees. To accurately realize the satellite attitude, precise starting and ending moments of these events (flip, ramp-up and ramp-down) are required and provided by the IDS ( ftp://ftp.ids-doris.org/pub/ids/satellites, Last access: 14 November 2019). A schematic illustration of the nominal yaw steering model is shown in Figure 2.

**Table 1.** Regimes of the nominal yaw steering model of Jason satellites together with the approximate duration of the different regimes. The values in the parentheses correspond to the period with $\beta'_{\mathrm{ramp}} = 30°$.

| Yaw Regime | Occurrence | Description | | Approximate Duration | | |
|---|---|---|---|---|---|---|
| | | | | Jason-1 | Jason-2 | Jason-3 |
| Sinusoidal | $|\beta'| > 15°$ | Yaw sinusoidal law | | 50 days | 50 days (4–93 days) | 50 days (24–48 days) |
| Fixed | $|\beta'| < 15°$ | Yaw = 0°, if $\beta' > 0°$ <br> Yaw = 180°, if $\beta' < 0°$ | | 11 days | 11 days (23 days) | 11 days (23 days) |
| Ramp-up | $|\beta'| \geq 15°$ | Yaw fixed to sinusoidal | | 120 s | 95 s (130 s) | 95 s (130 s) |
| Ramp-down | $|\beta'| \leq 15°$ | Yaw sinusoidal to fixed | | 120 s | 95 s (130 s) | 95 s (130 s) |
| Yaw flip | $|\beta'| \approx 0°$ | Yaw = 0°, if $\beta' > 0°$ <br> Yaw = 180°, if $\beta' < 0°$ | | 19 min | 10 min | 10 min |

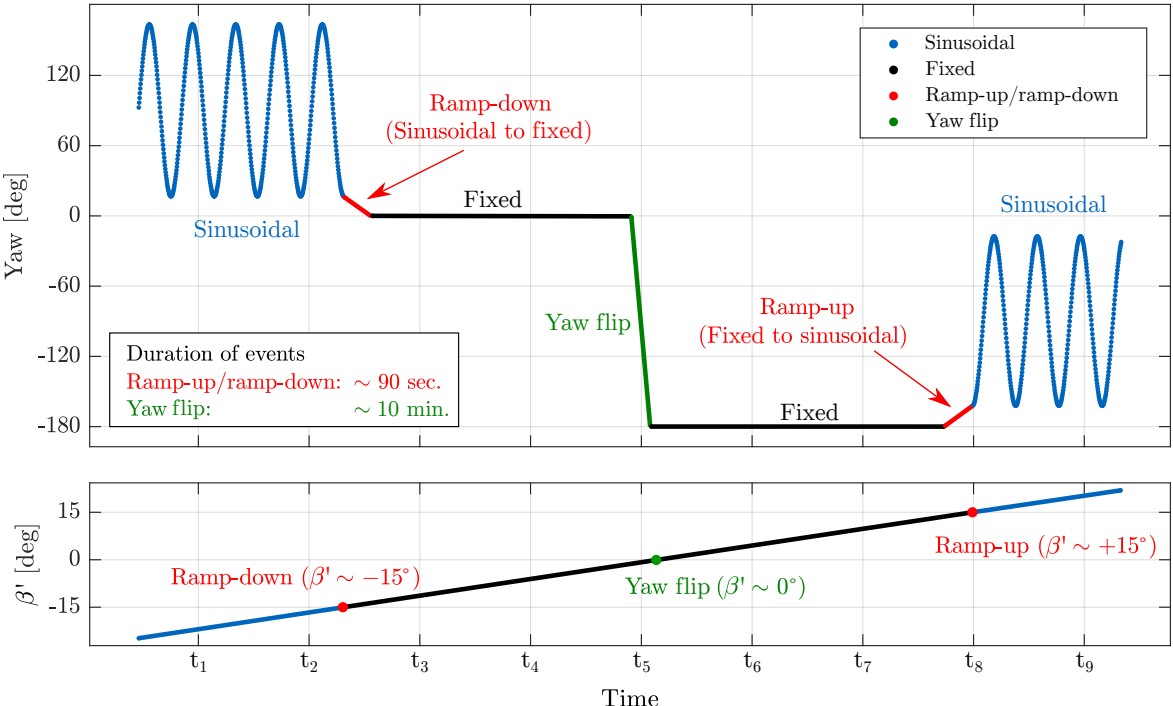

**Figure 2.** Principle of the nominal yaw steering model of the TOPEX/Poseidon satellite and all Jason satellites (top) depending on the angle $\beta'$ (bottom). Time instants are schematic.

### 2.1.2. Observation-Based Attitude and Its Processing

Observations which are used for the realization of the satellite attitude comprise quaternions of the spacecraft body and angular positions of the solar arrays. For Jason-1/-2/-3, they are provided

by CNES and are available via NASA's CDDIS (Crustal Dynamics Data Information System; ftp: //cddis.gsfc.nasa.gov/doris/ancillary/quaternions/, Last access: 14 November 2019). These data are given in separate daily satellite attitude files and solar panel angle files which individually cover a time span of 28 h (overlap of two hours between two consecutive files). In this work, Jason-1 data are used in the period from December 2001 until June 2013 (entire satellite mission), Jason-2 data from June 2008 until January 2019 and Jason-3 data from February 2016 until January 2019. The quaternion files provide the four quaternion elements ($q_s$, $q_x$, $q_y$, $q_z$) of the spacecraft and represent the satellite attitude with respect to the GCRS at epoch J2000. The solar panel angle files contain the commanded angular positions (from the ground control station) given, in radians, of the left and the right solar arrays. The difference between the commanded and measured panel positions is less than $1°$. Both data sets are given in Coordinated Universal Time (UTC) with a temporal resolution of approximately 32 s (the detailed format description is given by Ferrage and Guinle [16]). The angular differences of the observed and nominal roll and pitch components are up to $±0.1°$, the yaw differences are about $±2°$ (Figure 3). The observed and nominal solar panel angles differ up to $±15°$. For all components, the largest differences occur during the yaw flip and in the sinusoidal regime at small $β'$ angles. After switching the $β'_{ramp}$ angle from $15°$ to $30°$ on 12 August 2017 (6420 days since J2000.0), the differences between the observation-based and nominal solar panel angles decreased (Figure 3).

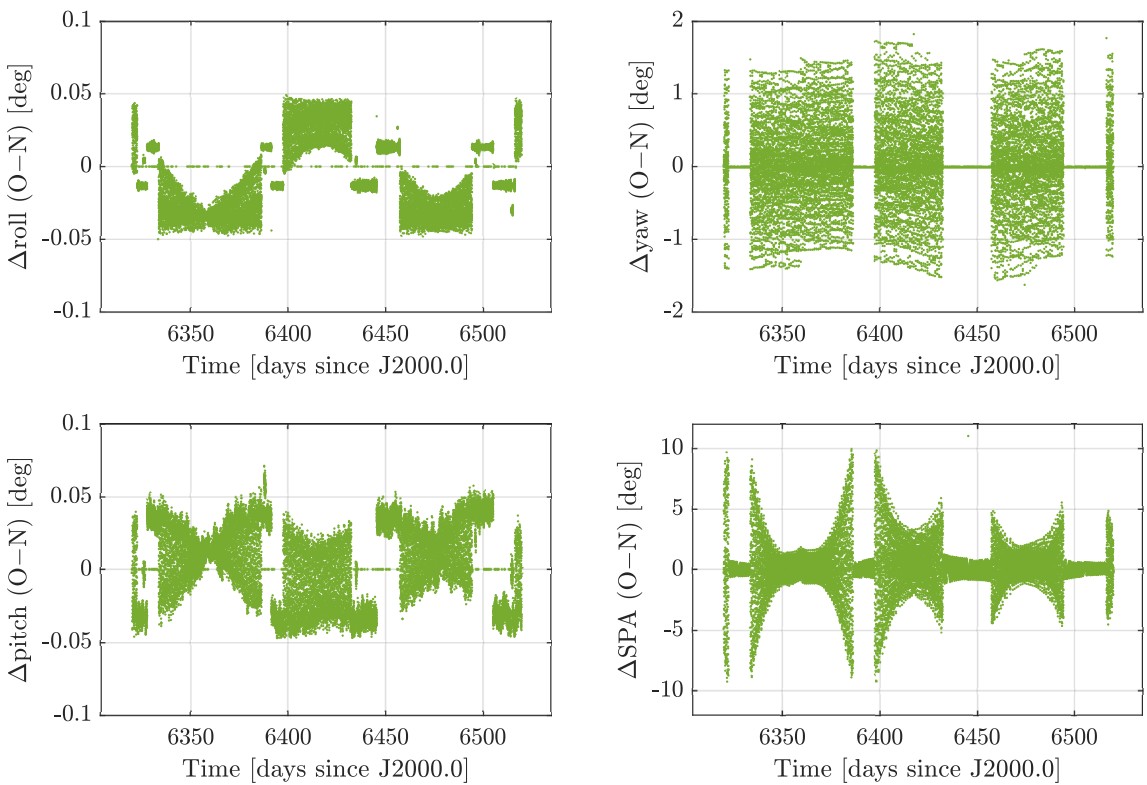

**Figure 3.** Differences between the observed (O) and the nominal (N) satellite attitude and solar panel angles in the RPY frame for Jason-3.

The following procedure for preprocessing of attitude data has been developed at DGFI-TUM. For an exact computation of the non-conservative perturbing forces, a complete set of six parameters, four quaternion elements and two solar panel rotation angles, must be given at each (SLR) measurement epoch and each orbit integration epoch. Hence, solar panel angles have to be interpolated at epochs that only occur in the quaternion files and vice versa. The final products of the preprocessing strategy applied at DGFI-TUM are attitude files aligned to Global Positioning System (GPS) weeks which cover a total period of nine days from Saturday $0^h$ UTC.

Moreover, during the preprocessing at DGFI-TUM, the following quaternion and/or solar panel angle entries are eliminated: (i) entries with duplicated epochs of the consecutive overlapping files, (ii) entries with zero values of quaternions and solar panel angles occurring at the beginning of the satellite mission and at the initialization of the quaternion and solar panel angle determination as well as in the case of sensor issues or safehold mode and (iii) entries where the quaternion norm differs from 1 by more than $2 \cdot 10^{-6}$. Figure 4 shows an example of the resulting roll, pitch and yaw angles (right panel) of corrupted quaternions not fulfilling the later requirement (left panel).

After the elimination step, the data sets might still contain periods of an increased measurement sampling rate: every second or fraction of a second instead of the usual 32 s time interval. Based on two criteria, a sufficient sampling rate of 32 s for further data use and data storage reduction, these single-second periods are temporally resampled to 32 s intervals. To avoid interpolation errors, epochs within data gaps are not interpolated as shown in Figure 5a. The threshold for data gaps is set to 66 s allowing the interpolation of a missing epoch in intervals of the regular sampling rate of approximately 32 s. This time difference is rounded up and applied before and after the missing epoch. For cases, when quaternions or solar panel angles are given at different time instants, an interpolation of the respective data to identical time instants is performed. For solar panel angles, linear interpolation is used whereas attitude unit quaternions are interpolated using the Spherical Linear Quaternion Interpolation (SLERP, [17]) method which provides according to [17] an optimal quaternion interpolation.

To provide best possible interpolation results within resampled data intervals, the original high-sampled data are taken into account. Figure 5b illustrates the principle of interpolation in periods of high-sampled data (only the scalar part of the quaternions is plotted). Epoch $t_I$ is given in the solar panel angle data but is missing in the quaternion data. Thus, a quaternion interpolation is applied at epoch $t_I$. The red dot represents the erroneous interpolation result when the cyan-colored neighbours in the temporally resampled data are used. The use of the neighbours from the original, high sampled data (magenta) provides a significantly better interpolation result (green).

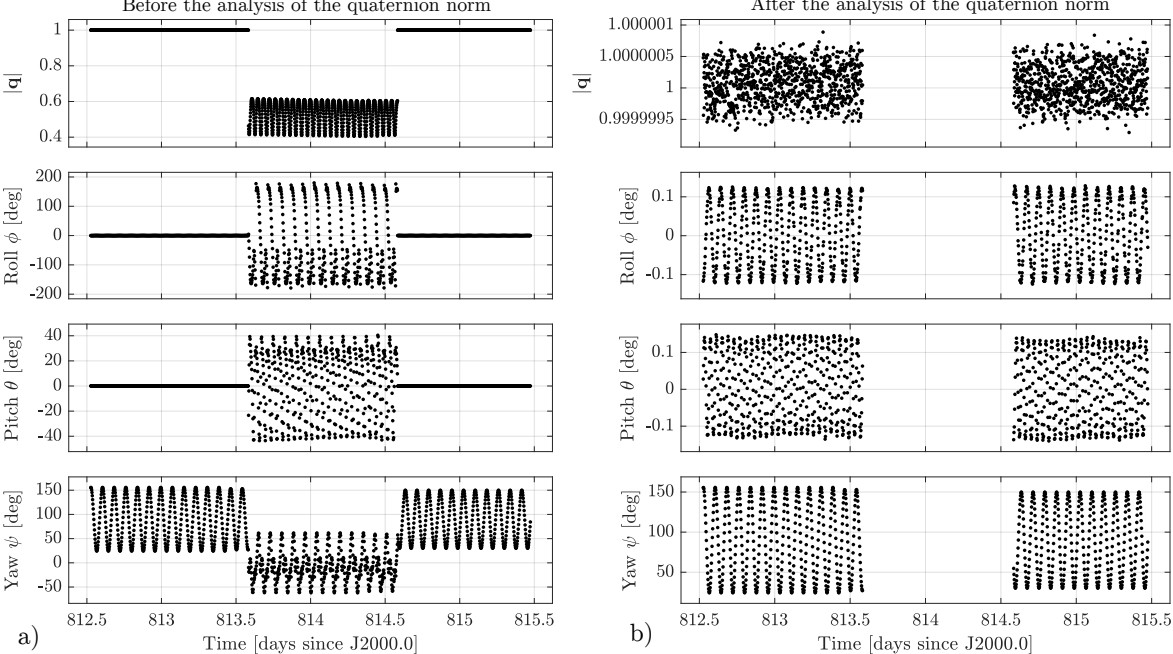

**Figure 4.** Analysis of the quaternion norm for the detection of outliers: (**a**) Non-normalized quaternions cause extremely oscillating attitude angles. (**b**) The corrected time series show the intended small roll and pitch angles.

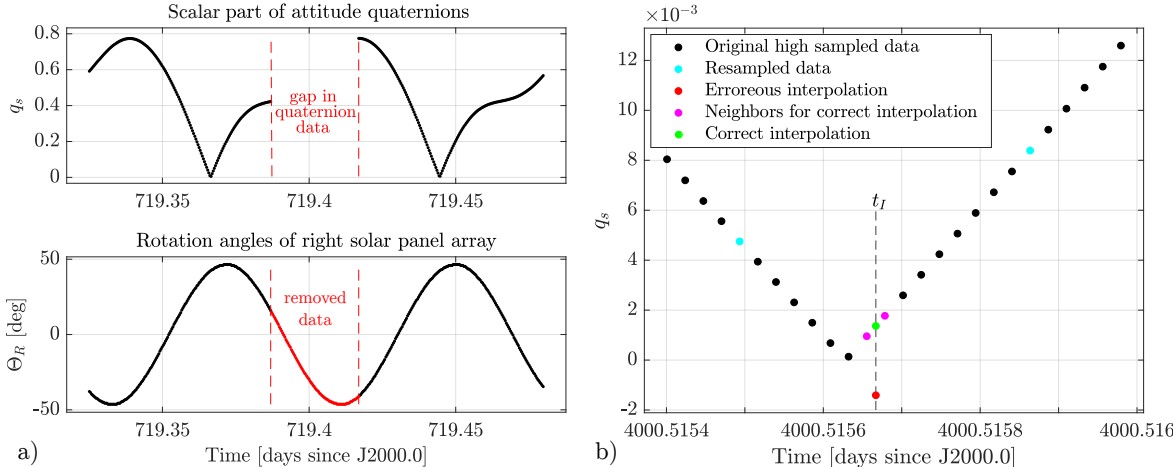

**Figure 5.** (**a**) Elimination of solar panel data in the case of a data gap in the quaternion time series. (**b**) Principle of interpolation in intervals of high sampled data.

## 2.2. Precise Orbit Determination

Using the DGFI-TUM Orbit and Geodetic parameter estimation Software (DOGS) we computed orbits of Jason-1/-2/-3 by modeling the satellite attitude in two forms: (i) by applying the nominal yaw steering model (named "nominal orbit" from here on) and (ii) by using observation-based attitude data (named "observation-based orbit" from here on). DOGS allows for the processing of observations of different geodetic space techniques and to combine equation systems at different levels of the Gauss-Markov-model [18]. For this study the following two libraries of DOGS were used: DOGS-OC (Orbit Computation library) and DOGS-CS (Combination and Solution library). For the POD, we use SLR observations available from the International Laser Ranging Service (ILRS, [19]) since this tracking technique is very sensitive to the radial orbit component which is directly related to the sea surface height measurements. In future studies, the impact also on DORIS-tracked orbits and combined SLR-/DORIS-orbits will be evaluated. Tables A1–A4 given in the appendix of this paper list the background models and settings used for processing SLR orbits. The usual orbit length for the Jason satellites is 3.5 days. Such an arc contains enough observations (approximately 1500) to compute a satellite orbit and estimate all required parameters. It also requires only a reasonable processing time (a few minutes). To avoid maneuvers in the orbit integration, shorter or longer arcs are computed, that is, orbits were truncated before a maneuver and restarted right after the end of a maneuver. SLR observations with an RMS greater than 12 cm are declared as outliers and eliminated from the orbit computation. Furthermore, SLR stations with an RMS over 10 cm are rejected as well. If an SLR measurement is within a data gap of attitude information, the nominal yaw steering model is used for the computation of the orbit using the SLR observations. The iteratively obtained satellite orbits cover the time intervals 13 January 2002 until 29 June 2013 (Jason-1), 20 July 2008 until 9 January 2019 (Jason-2), and 17 February 2016 until 9 January 2019. Within the POD of each arc, a set of 19 parameters is estimated:

- Initial state vector (once per arc),
- Solar radiation pressure scaling factor (once per arc),
- Earth's albedo scaling factor (once per arc),
- Atmospheric drag scaling factor (every 12 h),
- Sine and cosine coefficients of once per revolution empirical accelerations in along-track and cross-track direction (once per arc).

A-priori formal errors (constraints) of the scaling factors for solar radiation pressure, Earth's albedo and atmospheric drag were set to 0.1. The estimated once per revolution coefficients of the empirical accelerations in along-track and cross-track direction are unconstrained.

## 3. Results

In this section, several results are discussed which allow the evaluation of the impact of an observation-based attitude realization compared to a nominally modeled attitude. Section 3.1 discusses the SLR observation fits, Section 3.2 investigates the estimated orbital parameters, Section 3.3 comprises the external orbit validation, Section 3.4 presents the results of the analysis of station repeatabilities and finally, Section 3.5 studies the impact on derived sea level estimates.

### 3.1. SLR Observation Fits of Nominal and Observation-Based Orbits

The SLR observation fit is computed as the RMS value of the residuals between the measured range and the theoretical range (distance between the station reference point and the laser retro-reflector phase center computed from POD). The smaller this RMS value is, the better the computed range fits the measurements, that is, the reality. The arc-wise SLR observation fits obtained using the nominal yaw steering model for the attitude realization are shown in Figure 6. The values vary from the sub-centimeter up to almost 5.0 cm. The 60-day (about half the draconitic period of the Jason satellites) average of the observation fits is between 1.5 cm and 3.0 cm for all satellites. SLR observation fit outliers for Jason-1 larger than 4.0 cm are caused in this analysis by incorrectly modeled ramp events (e.g., due to an imprecise event information). These outliers vanish, that is, these arcs clearly improve, when using the observed attitude information. A few outliers above 4.0 cm for Jason-2 and Jason-3 do not correspond to a specific regime or event.

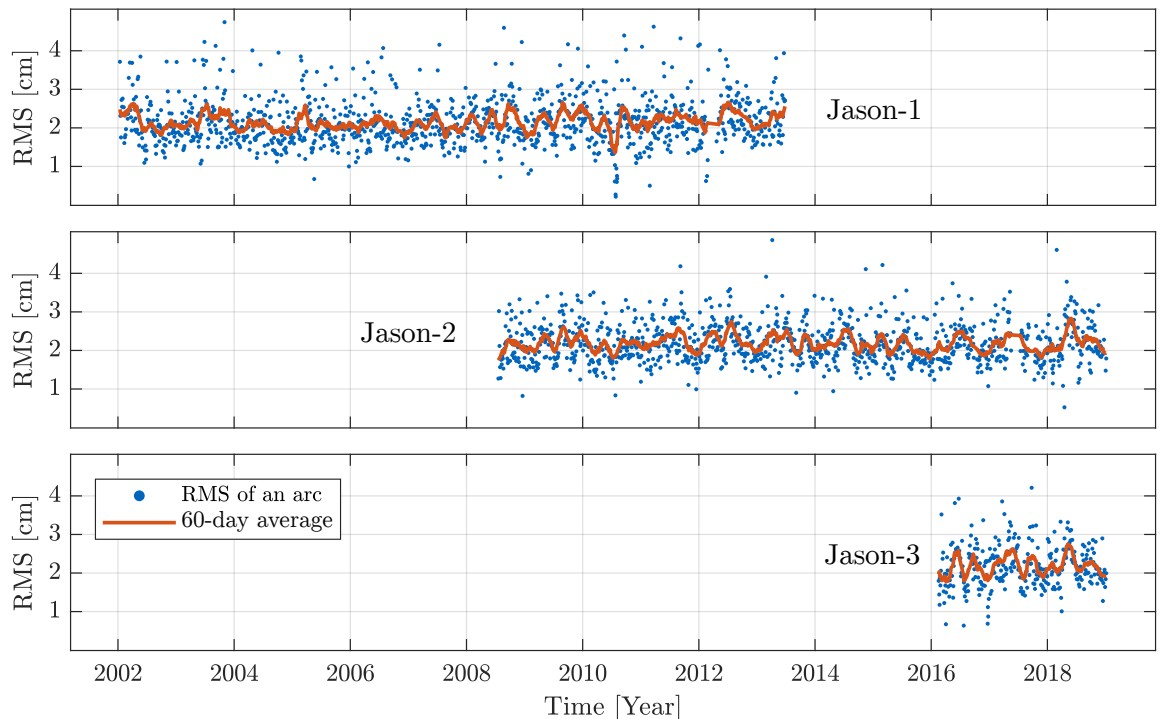

**Figure 6.** Arc-wise SLR observation fits of Jason-1/-2/-3 using the nominal yaw steering model.

In this analysis, we do not apply any biases published in the ILRS data handling file (see Table A3) because these biases were derived using SLR data to the spherical LAGEOS (LAser GEOdynamics Satellite) satellites and are not applicable to the Jason satellites. Moreover, all stations are weighted equally in the analysis. Very short arcs (24 h up to 48 h, potentially between satellite maneuvers) have less SLR observations which results in an observation fit smaller than 1.0 cm. This is due to the more flexible behavior of the orbit, since the satellite orbit is less constrained in the case of less observations during the orbit adjustment. An overview of the used number of observations and the identified and excluded outliers is given in Table 2.

**Table 2.** Results using nominal and observation-based attitude realizations. Shown are the amount of used observations per arc (left columns), the averaged SLR observation fit (middle columns) and the overall mission improvement (right columns). For the arc-wise SLR observation fit, please see Figure 6.

| Satellite | Observations per arc | | Averaged SLR obs. fit (cm) | | Improvement (N → O) | |
|---|---|---|---|---|---|---|
| | Mean Used | Mean Excluded | Nominal (N) | Observed (O) | SLR obs. fit | No. of arcs |
| Jason-1 | 1457 | 3.26% | 2.149 | 2.021 | 5.93% | 73.92% |
| Jason-2 | 1663 | 2.41% | 2.187 | 2.006 | 8.27% | 84.65% |
| Jason-3 | 1470 | 0.66% | 2.175 | 2.077 | 4.51% | 75.57% |

For all three satellites, most arcs have a smaller observation fit when realizing the satellite orientation by the observed attitude information. The relative change of the SLR observation fit between both attitude realizations is given in Figure 7. As shown in this figure, a reduction of the observation fits indicates an improvement of the computed orbit (marked in blue) while an increase indicates a degradation (marked in red). According to Table 2, 74% of the arcs of Jason-1, 84% of Jason-2 and 75% of Jason-3 are better compared to the orbits computed with the nominal yaw steering model. The biggest relative improvement and the smallest average SLR observation fit are obtained for the Jason-2 mission. The overall mission observation fit decreases from 2.18 cm to 2.00 cm (−8.27%). For Jason-1, the average observation fit changes from 2.14 cm to 2.02 cm (−5.92%) while the one of Jason-3 decreases from 2.17 cm to 2.07 cm (−4.51%). Regime/event-specific statistics of the SLR observation fits are shown in Table 3. Jason-3 has the smallest improvements and the biggest average SLR observation fit of the Jason satellites. This effect might be caused by errors in the background models, which are used to compute the perturbations of the satellite or the displacements of station coordinates. The models were mainly published before 2014 and generated with observation data obtained until the publication year (see Tables A1 and A2). In the case of Jason-3 (satellite launch and start of the mission were in early 2016), these models only provide predictions. These models do not consider unforeseeable, unique or abrupt events, for example, a displacement of a station within the SLRF2014 (being an extension of the International Terrestrial Reference Frame realization, ITRF2014, for SLR stations) due to an earthquake close to an SLR station after 2016.

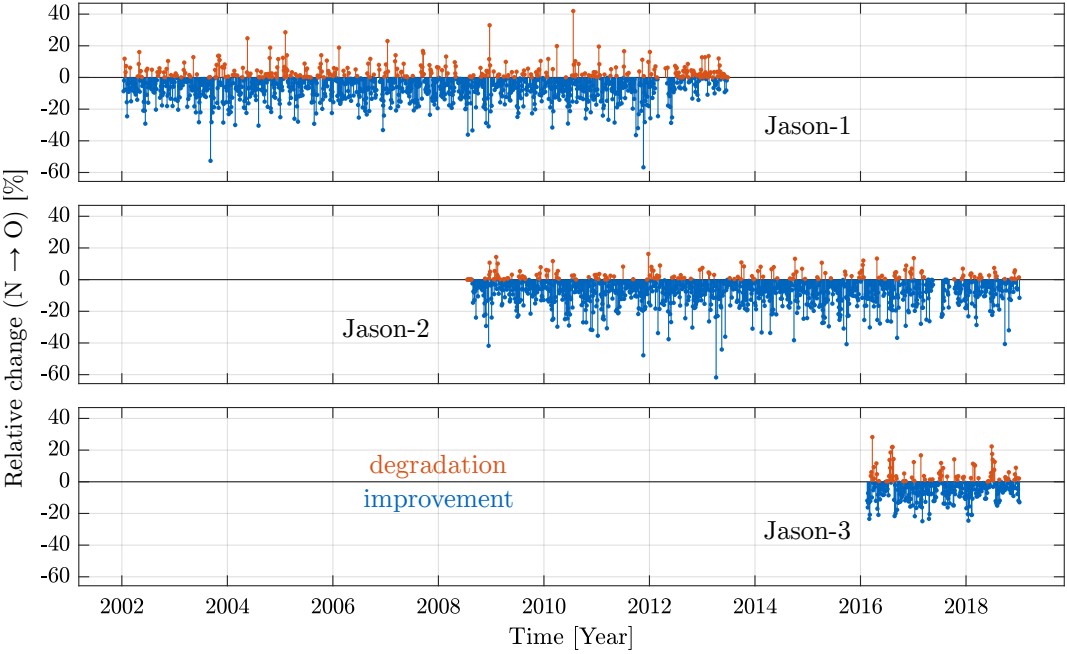

**Figure 7.** Relative change of the arc-wise SLR observation fit of Jason-1/-2/-3 from nominal to observation-based attitude realization.

**Table 3.** Statistics of the obtained SLR observation fits.

| Satellite | Yaw Regime | RMS Nominal (cm) | RMS Observed (cm) | Change N → O (%) | Number of Affected arcs |
|-----------|------------|------------------|-------------------|------------------|-------------------------|
| Jason-1 | Sinusoidal | 2.022 | 1.991 | −1.52 | 893 |
| | Fixed | 2.126 | 2.080 | −2.20 | 97 |
| | Ramp-up/down | 2.896 | 2.596 | −10.38 | 137 |
| | Yaw flip | 2.260 | 2.345 | 3.73 | 69 |
| Jason-2 | Sinusoidal | 2.140 | 1.970 | −7.94 | 767 |
| | Fixed | 2.096 | 1.987 | −5.193 | 117 |
| | Ramp-up/down | 2.485 | 2.121 | −14.64 | 128 |
| | Yaw flip | 2.262 | 2.183 | −3.51 | 63 |
| Jason-3 | Sinusoidal | 2.139 | 2.041 | −4.56 | 200 |
| | Fixed | 2.071 | 1.983 | −4.25 | 53 |
| | Ramp-up/down | 2.344 | 2.190 | −6.59 | 36 |
| | Yaw flip | 2.552 | 2.529 | −0.89 | 18 |

## 3.2. Estimated Parameters Derived from Nominal and Observation-Based Orbits

Besides the initial state vector estimated each arc, other orbital parameters are obtained within a POD with DOGS-OC. Figure 8 illustrates mission averaged parameters such as a mean solar radiation pressure scaling factor, a mean Earth's albedo scaling factor as well as a mean atmospheric drag scaling factor together with four coefficients of empirical accelerations. The estimated values are shown for orbits based on the nominal yaw steering model (blue) and the observation-based attitude realization (red). Each parameter has an optimal target value which is represented by a reference polygon (black) in this figure. Estimated parameters being close to the reference polygon values indicate that the background models are capable of well explaining orbital variations. Parameters which differ from the reference polygon absorb deficiencies in the POD. The reader should keep in mind that correlations exist between the estimated orbital parameters which are neglected in this figure.

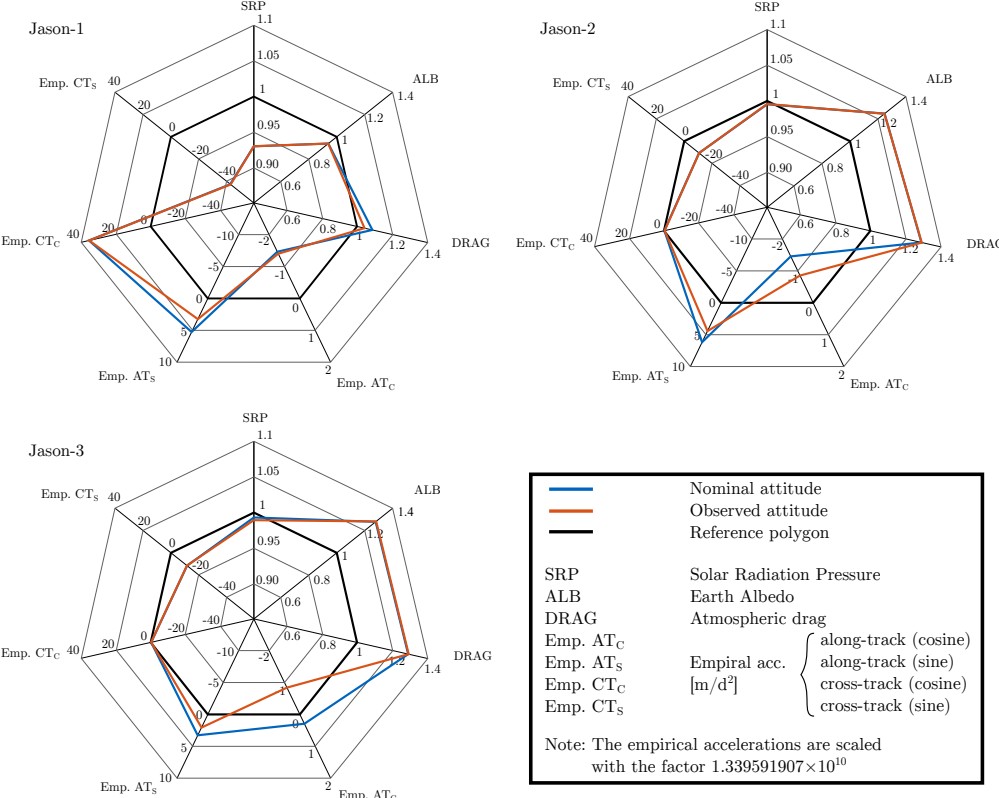

**Figure 8.** Averaged orbital estimated parameters within the POD of the Jason satellites. Estimates of the nominal approach (blue) and the observed attitude (red) which are closer to the reference polygon (black) indicate a better modeling of the (geo-)physical accelerations based on background models.

Figure 8 shows that the mean solar radiation pressure scaling coefficient for Jason-2 and Jason-3 is closer to 1 compared to Jason-1. This parameter absorbs modeling errors in all directions (radial, along-/cross-track) since the position of the Sun w.r.t. the satellite varies for all Jason satellites. The opposite situation is shown for both the Earth's albedo (absorbs radial modeling errors) and the atmospheric drag (absorbs along-track modeling errors) scaling factors. The empirical accelerations in along-track direction of Jason-3 show smaller offsets to zero than those of the other satellites. The cross-track empirical coefficients differ the most between the satellites. The cosine parameter is close to zero and the sine component is at about $-1.30 \cdot 10^{-9}$ m/s$^2$ for Jason-2 and Jason-3. For Jason-1, they are clearly bigger which indicates large unmodeled cross-track accelerations. It can be concluded that both attitude handling strategies result in similar orbit parameter estimates except the empirical along-track accelerations for all satellites. To a major extent, the along-track coefficients estimated for the observation-based attitude are closer to the reference polygon. This indicates a better along-track perturbation modeling based on geophysical background models due to a more reliable satellite attitude.

### 3.3. Orbit Comparison

For an external evaluation of the different attitude realization strategies, the SLR orbits are compared to orbits computed using SLR and DORIS observations at GFZ [7] and those computed using DORIS and GPS observations at CNES based on Geophysical Data Record (GDR, version E) standards ( ftp://ftp.ids-doris.org/pub/ids/data/POD_configuration_GDRE.pdf, Last access: 14 November 2019). Both (CNES and GFZ) orbits are computed using observation-based attitude data. We satellite-wise compare three different 1-month intervals at the beginning, middle and end of the each mission. For Jason-3 only two intervals are compared. Moreover, only CNES orbits are available for Jason-3 in this study. Figure 9 shows an example of Jason-1 orbit differences in the orbital reference system (along-track, cross-track, radial) for the time interval from 29 January 2012 until 15 February 2012. For all orbit differences, the smallest values occur in the radial component with a range of $\pm 5.0$ cm. The cross-track and along-track positions deviate between $\pm 10.0$ cm and $\pm 20.0$ cm, respectively. The figure shows that the DGFI-TUM SLR orbits are of a good quality compared to orbits computed at other institutions by a combination of geodetic space techniques.

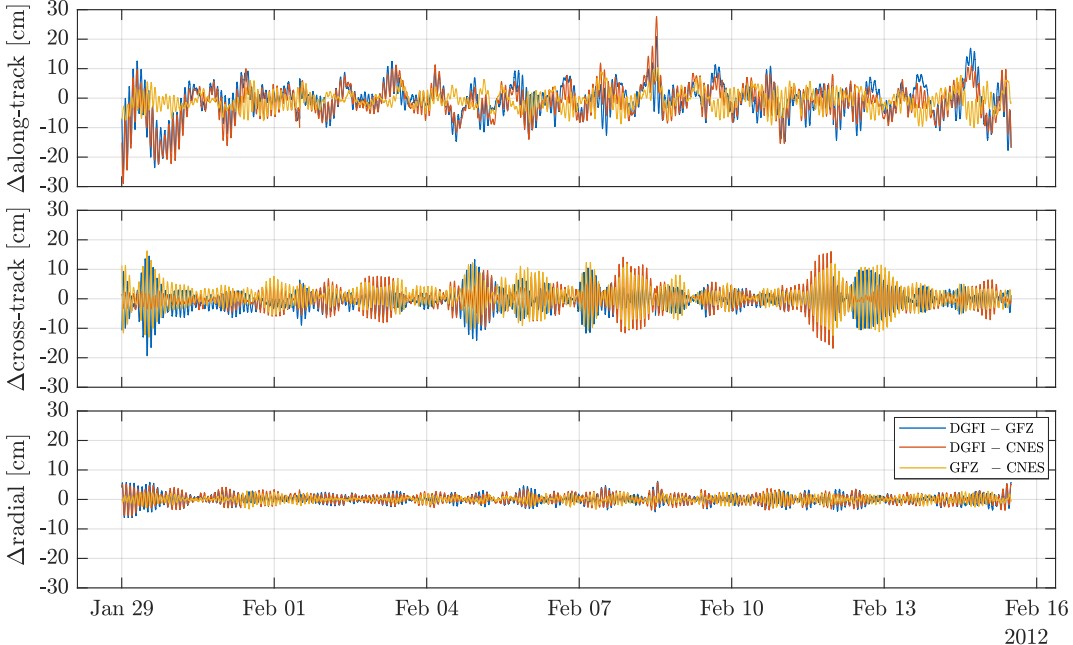

**Figure 9.** Jason-1 along-track, cross-track and radial orbit differences between DGFI-TUM, GFZ and CNES. The satellite was in sinusoidal regime the entire time interval.

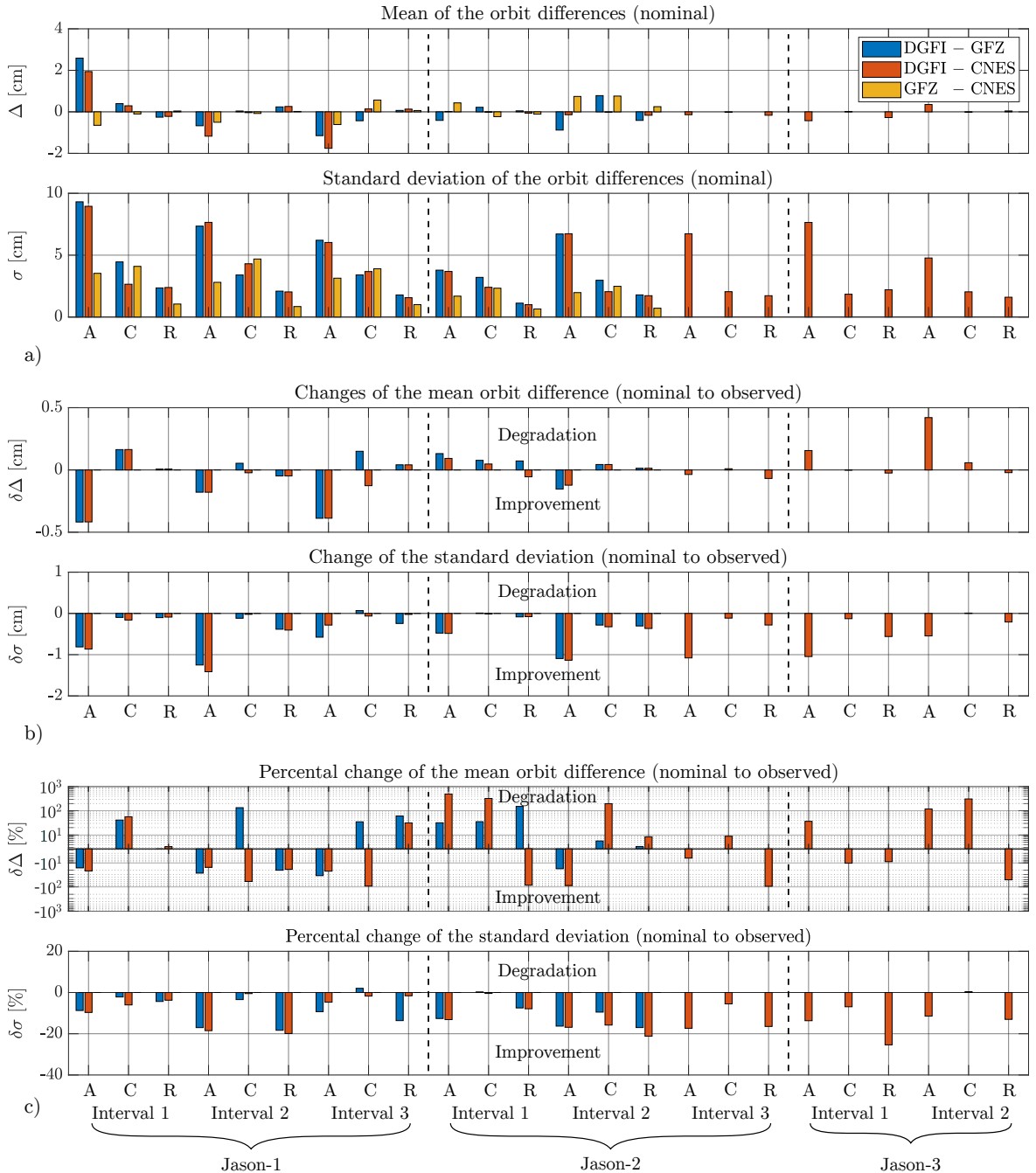

**Figure 10.** Results of the external orbit comparison of the orbit components (A = along-track, C = cross-track and R = radial) for the intervals used in this test. (**a**) Mean and standard deviation of the component-wise orbit differences between DGFI-TUM and GFZ and CNES. (**b**) Absolute and (**c**) relative changes between the nominal and observation-based attitude realization.

More detailed results of the external orbit comparison are illustrated in Figure 10. For each satellite-wise comparison interval, the component-wise determined orbit differences (A = along-track, C = cross-track, R = radial) are averaged. The mean values together with their standard deviations of the orbit differences obtained using the nominal yaw steering model are provided in subplot a). The orbit mean agrees well in most of the selected time intervals between the different solutions. Only Jason-1 orbits contain, mainly in the along-track direction, larger offsets up to 2.5 cm. The comparison of DGFI-TUM and CNES (red) Jason-2 orbits shows the smallest differences. The standard deviation of

the orbit offsets implies that the radial component between the orbits fits the best. On the contrary, the along-track direction contains the largest discrepancies.

The absolute and relative changes of the parameters from the nominal to the observation-based DGFI-TUM orbits are shown in Figure 10b,c, respectively. Negative values imply an improvement in the orbit comparison from the nominal yaw steering to the observed attitude. For GFZ and CNES, only one type of orbits is provided, hence, no orbit differences are determined. In general, the largest absolute changes are obtained in the along-track direction. The orbit difference of Jason-1 is about 0.2–0.45 cm smaller. Jason-3 orbit positions differ in the mean by about 0.5 cm more in the observation-based attitude approach. However, the standard deviation for almost all satellites and components is smaller for the observation-based attitude orbits compared to the nominal attitude orbits. For all satellites, the largest changes are obtained in the cross-track direction (up to 1.2 cm). Despite the changes at millimeter level, the other components improve by the same relative amount (bottom plot). The mean improvement of the standard deviation from the nominal orbits to the observation-based orbits is about 9.96%. This indicates that the use of attitude observations for precise orbit determination enhances the orbit accuracy and thus, the SLR orbits fit better to solutions computed using continuous tracking techniques like DORIS or GNSS. The change of the orbit difference mean does not show a consistent behavior. Note here the logarithmic scale of the plot. The biggest changes of up to 500% appear in Jason-2 intervals. This is due to a division by a small value of the initial mean orbit difference (Figure 10c).

*3.4. Effect on SLR Ground Station Coordinate Time Series*

In addition to orbital parameters, also geodetic parameters like arc-wise station coordinates were estimated and can be used for investigating the impact of the attitude realization. Exemplarily, the results of the spectral analysis of the SLR station coordinate time series of Graz, Greenbelt, Hartebeesthoek, Mt. Stromlo, Potsdam, Yarragadee and Zimmerwald are illustrated in Figure 11. These geophysical time series should be free from any orbit-specific signal such as the Jason draconitic period T $\approx$ 117 days and its harmonics 58 days (1/2 T) and 39 days (1/3 T) which are caused by an inaccurate attitude realization. Therefore, the amplitudes at these frequencies are estimated. The rows in the figure represent the station components observed by the Jason satellites (e.g., N1 is the station's North component derived from Jason-1 observations). Colored elements indicate a draconitic signal in the station time series. White-marked cells represent station components that do not contain any of the above-mentioned periods. The specified periods occur in 27% (17 of 63 for T), 35% (1/2 T) and 19% (1/3 T) of the station components. The contribution of affected coordinates is not homogeneous. The draconitic period occurs mainly in the North direction of the station coordinates while draconitic harmonics are present in all components. About 30% of the Jason-1 (N1 + E1 + U1), 25% of the Jason-2 and 45% of the Jason-3 station coordinate time series are corrupted by draconitc signals. The colors define the relative change between the attitude approaches. For blue-marked components, smaller amplitudes are found for the observation-based attitude data. This means that less or smaller orbit deficiencies propagated into the obtained station coordinate time series. The improvements are up to 40%, however, some directions show slightly increased amplitudes with a maximum of approximately 40% in the Up coordinate of station Zimmerwald. The average amplitude reduction of each period is 29% (T), 19% (1/2 T) and 7% (1/3 T).

The empirical quantities of the least square adjustment derived from the estimation of weekly SLR station coordinates are provided in Table 4. All three parameters, namely the a posteriori variance factor $\hat{\sigma}_0^2$, the sum of the squared corrections $v^T P v$ and the sum of the squared residuals $l^T P l$, improve from the nominal to the observation-based orbits by about 10% for Jason-1 and Jason-3 and almost 20% for Jason-2. Thereby, $v$ is the vector of corrections to the observations, $l$ is the vector (O-C) and the matrix P contains the inverse variances of the observations on the main diagonal. Smaller values of the before mentioned quantities imply a better fitting of the functional model within the estimation

process. Additionally, smaller corrections between the observed and computed SLR range (O-C) and the resulting residuals support this statement.

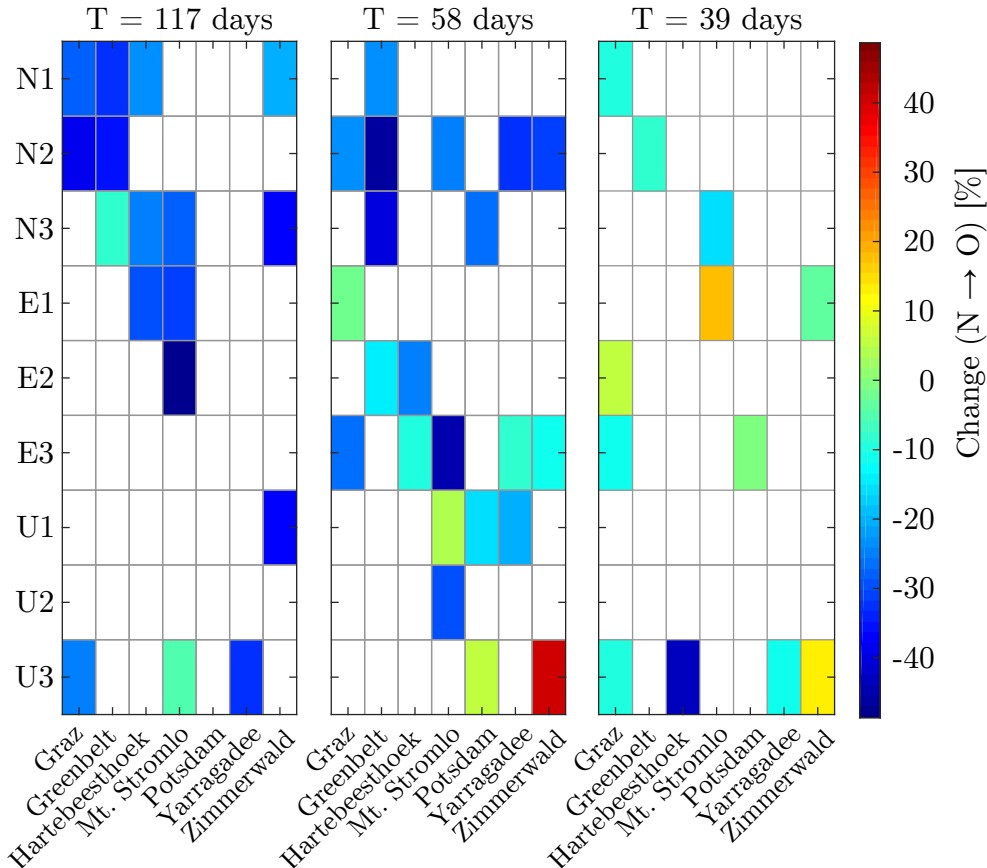

**Figure 11.** Relative change of the Jason draconitic (117.53 days) amplitudes within the estimated SLR station coordinate time series (N = North, E = East, U = Up).

**Table 4.** Changes of the a posteriori variance factor $\hat{\sigma}_0^2$, the sum of the squared corrections $v^T P v$ and the sum of the squared residuals $l^T P l$ obtained within an SLR station coordinate estimation based on either the nominal (N) or the observation-based (O) attitude orbits.

|  | **Jason-1** | | | **Jason-2** | | | **Jason-3** | | |
|---|---|---|---|---|---|---|---|---|---|
|  | **N** | **O** | **N → O** | **N** | **O** | **N → O** | **N** | **O** | **N → O** |
| $\hat{\sigma}_0^2$ | 2.8885 | 2.5451 | −11.89% | 2.9733 | 2.4120 | −18.88% | 2.5395 | 2.2887 | −9.88% |
| $v^T P v$ | 16069 | 14388 | −10.46% | 19165 | 15875 | −17.17% | 16655 | 15359 | −7.78% |
| $l^T P l$ | 8629 | 7671 | −11.10% | 10244 | 8344 | −18.54% | 7612 | 6866 | −9.80% |

These results prove that the observation-based attitude realization benefits the accurate estimation of SLR station coordinates. The reduction of orbit depending periods in the station position time series allows a more reliable geophysical interpretation of the station repeatability.

*3.5. Effect on Sea Level Estimates*

Within this section, two different altimetry results are compared based on two sets of sea surface heights (SSH): one computed using the nominal yaw steering based orbit, the other using the observation-based orbit solution. For this purpose, only SLR observations during the Jason-1/-2/-3 "nominal orbit" period are used ( https://directory.eoportal.org/web/eoportal/satellite-missions/j, Last access: 14 November 2019). In a first step, single-satellite SSH crossover differences with a maximum time difference of 10 days in open ocean areas are analyzed. Then, the mean and standard

deviations within the repeat cycles of Jason (about 10 days) are analyzed. For all three missions, a clear reduction in both parameters can be seen (Table 5). The mean of the absolute crossover differences is reduced by 6%, 15%, and 16% for Jason-1/-2/-3, respectively, when observation-based orbits are used. The observation-based orbits show overall mean crossover differences between 4.8 mm (for Jason-2) and 6.3 mm (for Jason-3). When comparing these values with other analyses, it is important to keep in mind that these are SLR-only orbits and the numbers will be reduced for most of the combined orbit products.

**Table 5.** Average statistic of the altimetry analyses of the nominal orbit (N), observation-based attitude orbit (O), and its differences for all three Jason missions on nominal ground tracks, that is, without interleaved and geodetic mission phases.

| | Mean Abs Crossover (mm) | Std Crossover (mm) | Std Rad Error (mm) | Mean GCE (mm) | Std GCE (mm) |
|---|---|---|---|---|---|
| **Jason-1** | | | | | |
| N | 5.84 | 61.56 | 22.98 | −0.171 | 5.550 |
| O | 5.49 | 60.53 | 22.12 | −0.170 | 5.582 |
| O-N | −0.35 | −1.03 | −0.86 | 0.001 | 0.032 |
| improvement [%] | 6.0 | 1.7 | 3.7 | 0.6 | −0.6 |
| **Jason-2** | | | | | |
| N | 5.69 | 59.97 | 16.93 | 0.081 | 2.340 |
| O | 4.83 | 58.30 | 15.25 | 0.070 | 2.300 |
| O-N | −0.86 | −1.67 | −1.68 | −0.011 | −0.040 |
| improvement [%] | 15.1 | 2.8 | 9.9 | 13.6 | 1.7 |
| **Jason-3** | | | | | |
| N | 7.48 | 59.08 | 15.62 | 0.357 | 3.818 |
| O | 6.26 | 58.33 | 14.51 | 0.183 | 3.490 |
| O-N | −1.22 | −0.75 | −1.11 | −0.174 | −0.328 |
| improvement [%] | 16.3 | 1.3 | 7.0 | 47.6 | 8.6 |

More important is the fact that the standard deviation of crossover differences within one repeat cycle (i.e., over the whole globe) is also clearly reduced in most cases. The percentage of cycles, in which the observation-based orbit can improve the results, is 88% in case of Jason-1, 87% for Jason-3, and 96% for Jason-2. The averaged differences are below 2.0 mm, however, some cycles show improvements of several centimeters as can be seen in Figure 12.

In addition to the single-satellite crossover analysis, a multi-mission crossover analysis (MMXO) as described by [20] has been performed based on the two different orbit solutions. This approach is useful to analyze the consistency between different missions and to identify systematic behaviors in single missions (e.g., [21,22]). This method has already been successfully used for investigating the effect of various parameters and background models used for POD, especially the impact of different gravity field solutions ([5]), atmospheric-ocean de-aliasing products [23], and reference frame realizations ([7,24]). For this study, four missions are used, namely, Jason-1/-2/-3 and SARAL for the time period between July 2008 and May 2017. Single- as well as dual-satellite crossover differences up to a time difference of 2 days are used. The main results of MMXO are estimated time series of radial errors for all missions involved in the analysis. These are used to study auto-correlation and amplitude spectrum as well as geographically correlated errors.

The radial errors computed based on the nominal attitude orbits show a larger scatter than those computed using observation-based attitude. The improvements are very similar to those yielded in the single-satellite crossover analysis (i.e., between one to two millimeters in average with extremes of up to three centimeters). The numbers for all three missions are summarized in Table 5 (std rad error). Figure 13 shows the empirical auto-covariance function of radial errors for all three Jason satellites for two orbit solutions. It can be clearly seen that for all three missions the use of an observation-based attitude orbit reduces the correlations after about 6752.5 s (i.e., after one orbit revolution) significantly.

The amplitude spectrum for Jason-2 also reveals the reduction in other frequencies, especially for a 59-day period and a 117-day period. This is not detectable for Jason-1 and Jason-3 probably due to their shorter time series (only overlapping time with Jason-2).

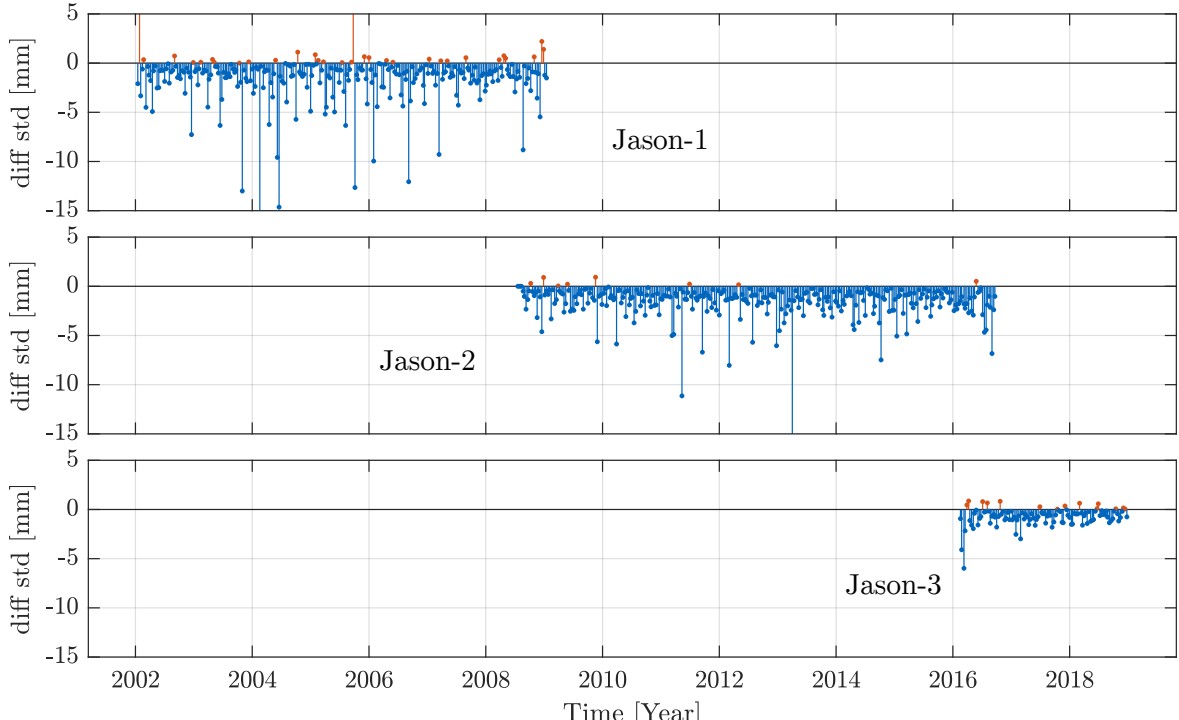

**Figure 12.** Single-satellite crossover differences (standard deviations per 10-day cycle) for the nominal orbit phases of the three Jason missions. Differences between solutions using quaternion-based orbits and nominal orbits (O-N). Negative differences mean improvements due to the observation-based attitude modeling.

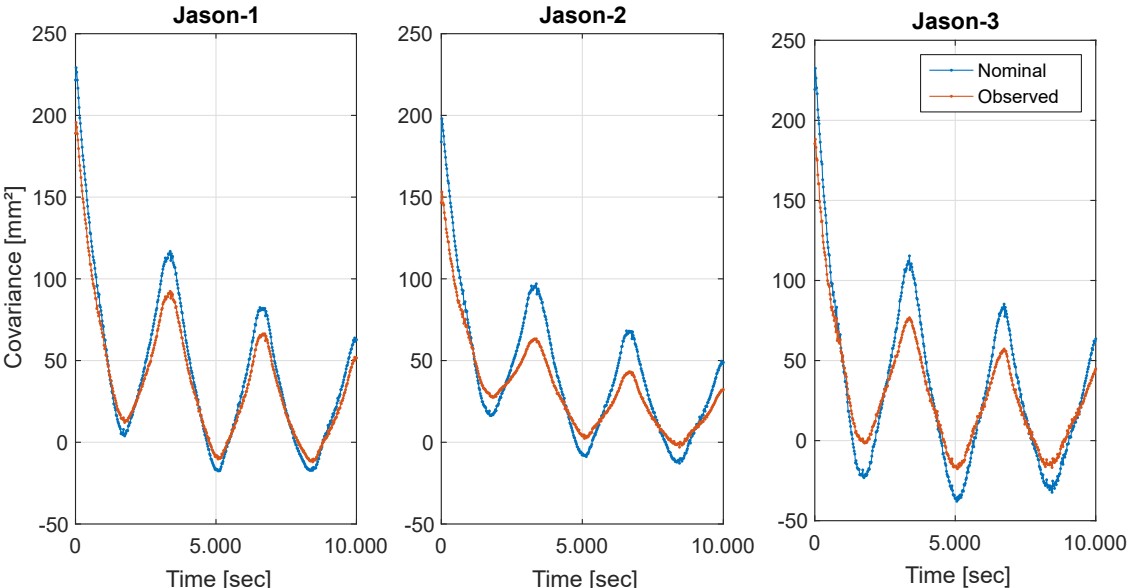

**Figure 13.** Empirical auto-covariance functions derived from time series of radial errors as estimated by MMXO. One can clearly see the reduction in overall variance as well as the reduction in correlation with the orbit revolution time.

MMXO also allows for analyzing geographical effects. For this purpose, geographically correlated mean errors (GCE) are computed based on the time series of radial errors (as described in [20]). Whereas the mean and the scatter of GCE are not significantly improved for Jason-1 and Jason-2, for Jason-3 clear improvements of 48% and 9% in both parameters are visible (see Table 5). Moreover, for this mission, a clear geographic pattern with amplitudes up to about 4 mm is visible when differencing the GCE computed with both orbit solutions (Figure 14, bottom right). The source of this pattern is still under investigation, but it has been confirmed by an independent calibration with Sentinel-3 and through direct comparison of radial orbit components.

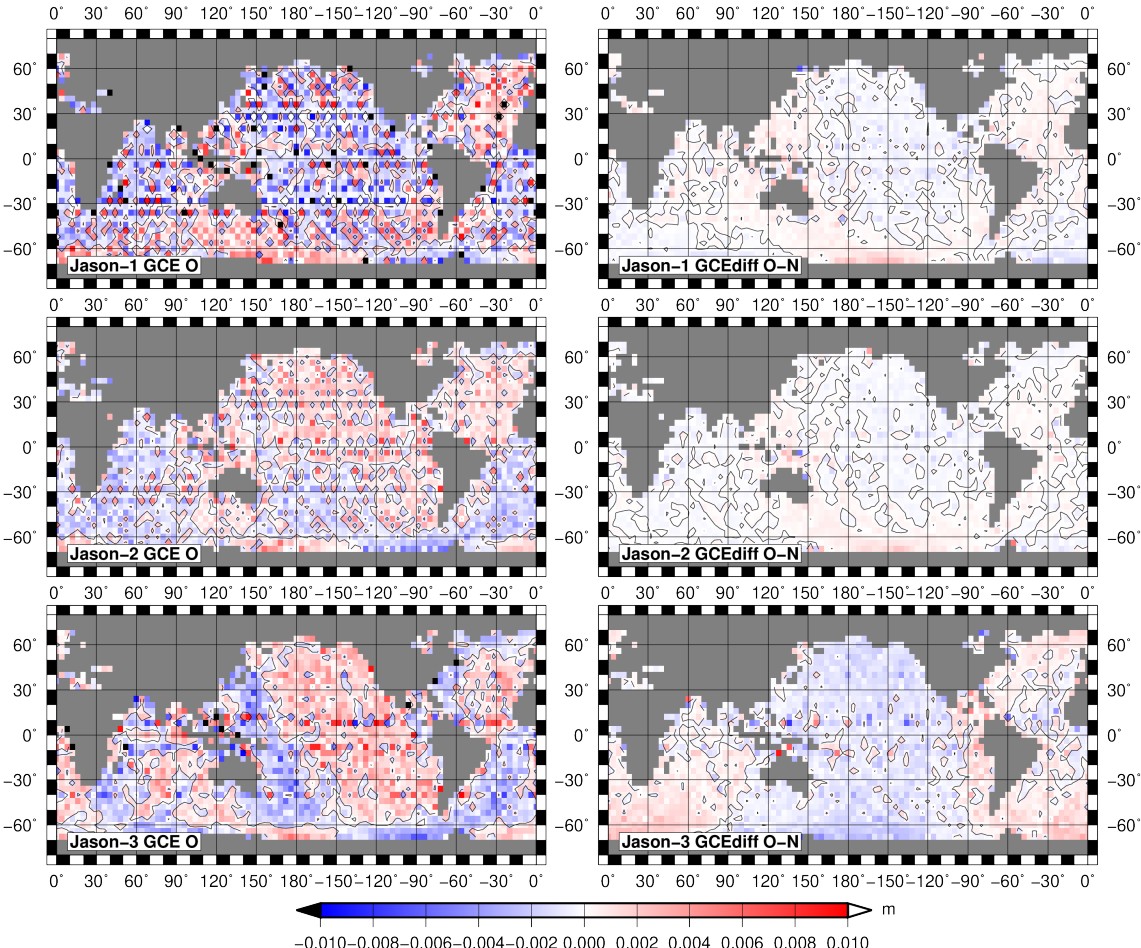

**Figure 14.** Geographically correlated error of all three Jason missions when using an observation-based orbit solution (left hand side) and its differences to GCE based on nominal orbits (right).

## 4. Discussion and Conclusions

In this paper, the differences in the attitude realization (observation-based compared to nominal yaw steering) in the POD of the Jason satellites are investigated. The impact of the different attitude realization strategies on satellite orbits is evaluated and consequences for derived geodetic parameters such as station repeatability and sea level estimates are quantified and discussed. The observation-based satellite attitude information comprise quaternions of the spacecraft body and rotation angles of the solar arrays. Nominally, the attitude can also be realized using a yaw steering model depending on the $\beta'$ angle. At DGFI-TUM, a preprocessing algorithm for the observation-based attitude data was developed. This algorithm comprises the elimination of outliers, a temporal resampling of the data and the optimal interpolation of missing data. Using SLR observations to all

three Jason satellites over a time interval of approximately 25 years, precise orbits are computed using both attitude approaches.

The analysis of the computed satellite orbits shows that using preprocessed observation-based attitude data reduces the averages of the arc-wise RMS fits of SLR observations by 5.93% (Jason-1), 8.27% (Jason-2) and 4.51% (Jason-3). The estimated orbital parameters such as solar radiation pressure scaling factors, Earth's albedo scaling factors, scaling factors for the thermospheric drag and coefficients of empirical accelerations obtained within a POD are analysed. Using observation-based attitude data significantly reduces the amplitudes of the parameter variations especially at draconitic periods.

The comparison of the DGFI-TUM orbits with external solutions of GFZ and CNES shows about 10% reduction of the standard deviations of the component-wise (along-track, cross-track, radial) orbit differences when using observation-based DGFI-TUM orbits. All orbit differences are of the same order of magnitude despite the fact that the GFZ and CNES orbits are based on a combination of geodetic space techniques, namely SLR and DORIS for GFZ and DORIS and GPS for CNES.

Furthermore, the impact of both orbit solutions on the estimated SLR station coordinates is investigated. Orbit specific periods (draconitic period and its harmonics) are reduced in the station coordinate time series using the observation-based satellite orbits. This allows a more reliable interpretation of geophysical signals in the station coordinate time series since modeling errors of the orbit do not cause them to deteriorate.

Our results prove that using observed attitude information is clearly better than utilizing models describing an intentional orientation. While the nominal model, in general, defines the long-wavelength behavior of the satellite orbit, observation-based data benefit the consideration of short-wave signals in the satellite orbit. The use of the actual attitude information allows a much more accurate determination of the perturbing forces. Hence, the computed satellite position is closer to the actual position and the differences between the observed and computed SLR observations are smaller. It has to be mentioned that the quaternions and solar panel orientation angles are considered to be error-free, that is, no standard deviations are considered.

The benefits for, for example, altimetry products are twofold. First, an improved satellite attitude corrects the position of the onboard altimeter in the GCRS. This implies an improvement of the measured sea surface height. Additionally, the pointing and consequently the georeferencing of the altimeter measurement footprint is more precise. Our single-satellite altimetry crossover analysis shows that the mean of absolute crossover differences is reduced by 6% for Jason-1, 15% for Jason-2, and 16% for Jason-3 when observation-based orbits are used. Also radial and geographically correlated mean errors are reduced. The potentially enhanced altimetry measurements can be used for computation of such products as sea surface topography, significant wave heights, ocean tides as well as global and regional sea level products. Using corrected measurements should increase the reliability of the mean observed global and local sea level change.

In this paper, only SLR observations are used for the computation of satellite orbits. In future investigations, a combination of geodetic tracking techniques for POD should be evaluated. DORIS and GNSS provide a considerably large and continuous number of satellite observations. Thus, the benefit of using attitude observation data in combination with these tracking techniques should be compared to the SLR-only results. To quantify the enhancement of the observation-based attitude approach at different orbit geometries, the investigation should be extended for other satellite missions. Therefore, the preprocessed observation data and the experience in data processing might be useful for other institutes.

The implementation of the observation-based attitude realization for GNSS and DORIS satellites is a currently discussed topic in recent workshops of the IDS and the International GNSS Service ( https://ids-doris.org/ids/reports-mails/meeting-presentations.html#ids-awg-04-2019, Last access: 14 November 2019 and http://www.acc.igs.org/workshop2019.html, Last access: 14 November 2019). This illustrates the contemporary objective of this study.

The improved orbits might also benefit other scientific disciplines, too. The corrected time series of station coordinates can be used to improve geophysical models. Moreover, beneficial or adverse effects on the Terrestrial Reference Frame can be further discussed. Based on the findings of the prior described investigation, it can be clearly stated that attitude observations help to further improve the POD of near-Earth satellites. Moreover, in this study, it was shown that an improved satellite attitude handling not only exceeds an improvement of the POD but further beneficially affects derived geodetic or geophysical parameters.

The preprocessed observation-based attitude data of Jason-1/-2/-3 will be made available on request at DGFI-TUM (contact responsible author of this manuscript).

**Author Contributions:** M.B. conceptualized the study; M.B. and J.Z. elaborated the algorithms of nominal and observation-based attitude use in DOGS-OC; J.Z. developed the algorithm for preprocessing of the observation-based attitude data; J.Z. and S.R. derived all satellite orbits; D.D. evaluated the attitude realizations based on altimetry products; M.B., J.Z., S.R. and D.D. analyzed the obtained results and wrote the manuscript. All authors have read and agreed to the published version of the manuscript.

**Funding:** This research received no external funding.

**Acknowledgments:** The authors thank CNES for providing Jason satellite attitude data via NASA CDDIS. SLR data of Jason satellites available from ILRS were used in this research. The authors are also grateful to IDS for making available Jason satellite specific information at its web pages. This publication is supported by the Technical University of Munich (TUM) in the framework of the Open Access Publishing Program. The authors acknowledge the contribution of the assistant editor of MDPI and the three anonymous reviewers for improving the readability of the paper.

**Conflicts of Interest:** The authors declare no conflict of interest.

## Appendix A

**Table A1.** Background force models for Jason satellite POD with DOGS-OC at DGFI-TUM.

| Forces | Model Used |
| --- | --- |
| Earth gravity field | EIGEN-6S model [25]<br>Static part: up to degree/order 120<br>Time variable part: up to degree/order 50 |
| Lunar gravity field | Up to degree/order 10 [26] |
| Third body effect | DE-421: Sun, Moon, Mercury, Venus, Mars, Jupiter, Saturn [27] |
| Solid Earth tides | International Earth Rotation and Reference Systems Service (IERS) Conventions 2010 [28] |
| Permanent tide | Conventional model (IERS Conventions 2010) |
| Ocean/atmospheric tides | EOT11a model [29] up to degree/order 30<br>+ BB2003 [30]<br>+ 62 admittance waves [28] |
| Solid Earth pole tide | IERS Conventions 2010 |
| Ocean pole tide | [31] |
| Non-tidal perturbations | Not applied |
| Solar radiation pressure | Constant radiation with eclipse modeling |
| Earth radiation pressure | albedo and infrared [32] |
| Atmospheric drag | JB2008 model [33]<br>+ External geomagnetic storm and solar flux indices |
| General relativistic correction | Schwarzschild, de Sitter Lense-Thirring (IERS Conventions 2010) |
| Thermal radiation | Applied [1] |

[1] https://ilrs.cddis.eosdis.nasa.gov/docs/Jason-1_specs.pdf, Last access: 14 November 2019.

**Table A2.** Background models for the computation of station coordinates during the POD in DOGS-OC.

| Station Background Models | Description |
| --- | --- |
| A priori station coordinates and velocities | SLRF2014 (ftp://cddis.nasa.gov/slr/products/resource/SLRF2014_POS+VEL_2030.0_171024.snx, version: 24 October 2017) |
| A priori Earth's Orientation Parameter (EOP) values | IERS EOP 14 C04 [1] with IERS 2010 daily/sub-daily corrections (sub-daily oceanic tide model [34]) |
| Precession/nutation | IAU2000A/IAU2006 model up to degree 10 [35] |
| Mean pole | Polynomial model (IERS 2010 Conventions) |
| Solid Earth tidal displacement | Anelastic model (IERS Conventions 2010) |
| Permanent tide | Conventional model (IERS Conventions 2010) |
| Ocean tidal displacement | EOT11a [29] |
| Atmospheric tidal displacement | S1/S2 tidal model [36] |
| Rotational deformation due to polar motion (solid Earth pole tide displacement) | Secular pole model (IERS Conventions 2010) |
| Ocean pole tide displacement | [31] |
| Non-tidal displacement (atmos- pheric, oceanic, hydrological) | Not applied |

[1] https://www.iers.org/IERS/EN/DataProducts/EarthOrientationData/eop.html, Last access: 14 November 2019.

**Table A3.** SLR measurement corrections applied in DOGS-OC for the computation of Jason orbits.

| SLR Measurement Corrections | Description |
| --- | --- |
| Tropospheric model for optical signals | Mendes-Pavlis with temperature correction [37] |
| Retro-reflector and timing instrument correction | Satellite-dependent CoM correction models [10] |
| Relativistic range correction | IERS Conventions 2010 |
| Station eccentricities | ILRS eccentricity file |
| Station discontinuities | ITRF2014 |
| Range and time biases | none |

**Table A4.** Settings for the orbit integration, observation handling as well as utilized satellite characteristics for the POD of Jason satellites.

| Orbit integration | |
| --- | --- |
| Reference frame of orbit integration | Geocentric Celestial Reference System (GCRS) |
| Integrator type | Predictor-corrector algorithm of Gauss-Jackson (7th order) |
| Integrator step size | 60 s |
| Integrator relative error bound | $10^{-9}$ |
| **Observations** | |
| Observation weight | 1 cm, no elevation dependent weighting applied |
| Elevation cut-off angle | 5° |
| LRA phase center offset | Constant range correction of 4.9 cm subtracted from the computed range |
| **Satellite characteristics** | |
| LRA optical center | [10] |
| Initial mass | [10] |
| Initial center of mass | [10] |
| Macro model | CNES box-wing model [16] |

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
