# Peer review of "Observation-Based Attitude Realization for Accurate Jason Satellite Orbits and Its Impact on Geodetic and Altimetry Results"

_remotesensing, doi:10.3390/rs12040682_

Round 1

Reviewer 1 Report

The paper developed a procedure for processing satellite attitude data and applied it to the POD of the Jason satellites. Different comparisons were performed to proof the attitude-observation based orbits are better than the nominal orbits. The conclusions are well supported by the results. So I suggest the manuscript accepted after a minor revision.

comments:

Add some review or references about the satellite attitude processing in Introduction. Figure 2 has no x-tick labels. Line 197-198: Can the author provide some concrete proof for the statement "An orbit fit over 4.0 cm can result from ....."? Line 201-202: Please explain why less SLR observations result in a better orbit fit. Line 325: "Jason-1" might be "Jason-2" in this sentence.

Reviewer 2 Report

The work would be worth to be published, but several amendments of the current manuscript are needed prior to publication in "remote sensing".

Comments, questions and corrections:
- p.2, l.29-31: "Altimetry satellites like such as ... are mounted and solar panels." => "Altimetry satellites such as ... and solar panels are mounted."
- p.2, l.51: "Section 2 provides information of ..." => "Section 2 provides information on ..."
- Introduction: There is no information how other groups/institutions are handling the attitude for the Jason or other satellites. Are there no additional references available about this? More information should be added in the introduction.
- p.2, l.59-61: "A detailed overview of the Jason satellite mission goals, ..." This sentence does not really fit here. It would better fit into the introduction.
- p.4, l.112: "...but can diver ..." => "... but can differ ..."
- p.4, Section 2.1: It would be of interest for the reader on how long the sinusoidal and fixed attitude regime normally are, approximate durations would be fine, e.g., hours, days, weeks or months.
- p.4, l.125: " ...in radiant..." => should this not be "...in radian..." ?
- p.4, l.126: You mention that the difference between the commanded (modeled?) and measured solar panel angle is less than 1 deg. How large are the differences between the nominal and measured attitude? Are there larger differences during one of the different attitude regimes or not?
- p.4/5, Section 2.2: What are you doing when a (SLR) measurement falls into the data gap? Are you switching to the nominal attitude law or do you reject the corresponding observation for POD? Please specifiy here in Section 2.2 or in Section 4.1.
- p.6, Figure 4b) "Neighours ..." => "Neighbours ..."
- p.7, l.181: This comment might not be diretly relevant for the results presented in this paper, but I am wondering which constraints you are using for the Earth's Albedo scaling factor? Have you heavily constrained it? This scaling factor is very much correlated to the scale of the system itself.
- p.7-9, Sections 4.1 + 4.2: Are there larger differences for intervals with sinusoidal attitude or any of the other attitude regimes? Can you give statistics for this? Or is there no significant difference between nominal and observed attitude in the different attitude regimes?
- p.9-12, Section 4.3: What are CNES and GFZ using for the attitude, the nominal model or the observations? Would be interesting.
Why do you only compare selected intervals and not the entire processed interval? Is there a reason for this?
- p.10, Figure 8: Which attitude was the satellite flying during this time interval? Only the sinusoidal attitude or also fixed ? If it is not only one attitude regime, the transition times would be interesting to highlight.
- p.13, l.313: "These results proof ..." => "These results prove ..." or "The results give proof ..."

- There are several very long sentences in the manuscript. This makes it a bit difficult to read. Please re-check the manuscript for these long sentences and re-phrase them into several sentences. It would be worth to do so.

Reviewer 3 Report

General comments

This paper should not be accepted in its present form in the MPDI Remote Sensing

The manuscript contains bewildering thoughts, which are not focused and specific. The abstract is not self-contained and explanatory. No abbreviations should be contained in the abstract. The manuscript contains generalities with no crystal clear and specific methodology as to what is done, why is done, and how it is done. The English language is not correctly used and the manuscript contains spelling, structure and grammar errors. I express my serious concern as regards to the numbers given to represent averages and uncertainties of Fig. 6 (and Fig 11). Diagrams contain a lot of outlying values and those are somehow related to yaw rotations of Jasons. Also, outliers are not quantified in this manuscript. It is preferable to see the RMS given in Fig. 6 as box plots as well  and in addition to Fig.6, to get an appreciation of what is going on there. The same should be done for Fig.8 and Fig.11 as well. The term “mean” as used in this manuscript refers to “averages”. Mean requires the whole statistical population, and this is not the case in this manuscript. Please correct. Also all “errors” have to be replaced by uncertainties, as none knows what the "true" value is to state that this parameter is an error. Please revise. The criterion of there-sigma in orbit computation involves sensitive estimation issues, and results may be shown as desired if no rules for its estimation are provided. Please provide explanation to the above. Also the way outliers were excluded from processing should be described clearly in the manuscript. I also express my concerns regarding the ground-truth values as computed by a 10 day difference for the determination of sea-surface height for cross-overs. This is too long period to support reliable values for testing your results.

Specifically:

The abstract is not crystal clear and specific. It has to be rewritten. For example, even in the first sentence of the abstract, there is comparison between “attitude observations” against “yaw” steering model only. Abbreviations in the abstract should be avoided. Abbreviations should be spelled out the first time they appear in the manuscript, but their definition should not be repeated throughout the manuscript several times as done in this case. For example, “GCRS” and “RMS” are defined three times in this manuscript. English language of this manuscript requires heavy editing. It contains mistakes, such as nouns are used as verbs (for example: These results proof that the observation-based (To proof= to protect something from being affected or damaged by something else, the same holds true for “effect: as used on page 3),”diver” by some degrees (page 4)…, sea level change, see, e.g., [1], start “tracking” cameras,  etc.). It is preferable to use “Draconic” instead of “Draconitic” (superfluous information). The word “exemplary” is used incorrectly. Also on page 7, please correct  “12-hourly”. Abbreviations, such as “SHM”, and “OPR” are defined, but never used in the manuscript. Also other abbreviations, such as SLRF2014 are given without an explanation or reference. Sentences are long and sometimes their meaning is lost. Figure 1 has to be redrawn, please. Angles, planes, reference system and axes are not clearly presented there. Also the “ecliptic” is not a line as shown in the same Fig.1. Sentences such as “The SLR orbit fit is computed as the RMS of the residuals between the SLR measurements and the integrated satellite orbit” and its  following sentence do not make sense. Please revise. Please provide supporting evidence for statements like “Moreover, all stations are weighted equally in the analysis.” This is injection of opinion. Please explain how “SLR fit” is accessed as given in Table 2. This is not given in the manuscript. Time units are not given in Fig.2. Please correct. Please clearly explain and provide justification as to how and why this value of “32 seconds” came up, as described on page 5 (None of this justification is given on Ref [14]). Never start a sentence with numbers. Please see page 12. The word “exemplarily” is not used correctly in the manuscript. Section 4.4 title seems to refer to SLR ground stations. Please rephrase. Please provide evidence and supporting statements why the “SLERP” interpolation is optimal, as stated on page 5. Also please provide supporting evidence why the arc length of Jason is 3.5 days. This is not given in the manuscript. Parameters in the least-squares adjustment processing (page 12) are given without any definition and or explanation as what these parameters mean.
